# Ventral hippocampal OLM cells control type 2 theta oscillations and response to predator odor

Sanja Mikulovic[1], Carlos Ernesto Restrepo[1], Samer Siwani[1], Pavol Bauer[2], Stefano Pupe[1,3], Adriano B. L. Tort [3], Klas Kullander [1] & Richardson N. Leão [1,3]

Dorsal and ventral hippocampus regions exert cognition and emotion-related functions, respectively. Since both regions display rhythmic activity, specific neural oscillatory pace-makers may underlie their functional dichotomy. Type 1 theta oscillations are independent of cholinergic transmission and are observed in the dorsal hippocampus during movement and exploration. In contrast, type 2 theta depends on acetylcholine and appears when animals are exposed to emotionally laden contexts such as a predator presence. Despite its involvement in emotions, type 2 theta has not been associated with the ventral hippocampus. Here, we show that optogenetic activation of oriens-lacunosum moleculare (OLM) interneurons in the ventral hippocampus drives type 2 theta. Moreover, we found that type 2 theta generation is associated with increased risk-taking behavior in response to predator odor. These results demonstrate that two theta oscillations subtypes originate in the two hippocampal regions that predominantly underlie either cognitive or emotion-related functions.

[1] Developmental Genetics, Department of Neuroscience, Uppsala University, Husarg 3, Uppsala 75234, Sweden. [2] Division of Scientific Computing, Department of Information Technology, Uppsala University, Box 337, Uppsala 75105, Sweden. [3] Brain Institute, Federal University of Rio Grande do Norte, Av. Nascimento de Castro 2155, Natal, RN 59056-450, Brazil. These authors contributed equally: Sanja Mikulovic, Carlos Ernesto Restrepo, Klas Kullander, Richardson N. Leão. Correspondence and requests for materials should be addressed to S.M. (email: sanja.mikulovic@neuro.uu.se) or to R.N.Lão. (email: richardson.leao@neuro.uu.se)

Theta oscillations are one of the strongest and most regular rhythms of the brain[1]. Hippocampal theta oscillations refer to a pair of rhythms with frequencies ranging from 4 to 12 Hz. Type 1 theta oscillations (theta1, 7–12 Hz) emerge during exploration and voluntary movement[1]. Type 2 theta oscillations (theta2, 4–9 Hz) have been linked to immobility and emotional states such as anxiety and innate responses to predator odor[2–5]. The two forms of theta oscillations were initially separated by their dependence on cholinergic drive and sensitivity to anesthetics (e.g., urethane): theta1 is atropine-resistant and sensitive to anesthetics whereas theta2 is atropine sensitive and resistant to anesthetics[2,3]. A recent study found a strong link between the activity of glutamatergic cells in the medial septum (MS) and theta1[4]. In contrast, stimulation of septal cholinergic cells has little effect on theta1 but increases the power and coherence of theta2 in anesthetized animals[6], implying the existence of distinct circuits behind theta1 and theta2.

Theta varies across different hippocampal regions. For example, movement-related theta1 power progressively decreases along the dorsoventral hippocampal axis[7–9], and the ventral hippocampus (vHipp) displays oscillations a slightly lower theta frequency during stress and fear-related behaviors[7,10,11]. Fear-associated theta oscillations are also synchronized in the hippocampus, medial prefrontal cortex, and amygdala[12]. Differences in vHipp and the dorsal hippocampus (dHipp) oscillatory patterns could also reflect cellular differences between these two subregions. Electrophysiological properties of CA1 pyramidal cells (PCs) differ between the dHipp and vHipp[9,13]. Ventral CA1 (vCA1) PCs display higher excitability and stronger hyperpolarization-activated currents[13,14], and apical PC dendrites in the vHipp show enhanced backpropagation, theta resonance, and lower temporal summation of synaptic inputs than in the dHipp[14,15]. Afferent inputs also differ: the dHipp receives preferentially spatial and visual information, whereas the vHipp receives olfactory inputs and is highly connected with emotion-related circuits[16,17]. Hence, it is plausible that the dHipp and vHipp may produce distinct oscillatory patterns.

Here, we investigated the role of the CA1 oriens lacunosum-moleculare (OLM) interneuron subtype specifically expressing the nicotinic acetylcholine receptor $\alpha2$ subunit (Chrna2; OLM$^{\alpha2}$ cells)[18,19] in generating local field potential (LFP) oscillations. We show that optogenetic activation of OLM$^{\alpha2}$ cells in the vHipp is sufficient to induce theta2, while optogenetic inhibition decreases naturally occurring theta2 in both anesthetized and freely behaving animals. Finally, we find that optogenetic generation of theta2 leads to a significant increase in risk-taking in the predator odor test, while its inhibition decreases this behavior.

## Results

### OLM$^{\alpha2}$ stimulation induces theta2 in anesthetized animals.
We have previously shown that the Chrna2-Cre transgenic mouse line reliably marks OLM cells in the CA1 hippocampal region[18] and that the vast majority of OLM$^{\alpha2}$ cells are confined to the vCA1[19]. Here, we further investigated the distribution of OLM$^{\alpha2}$ cells using counting in slices and three-dimensional analysis. We found that the majority of OLM$^{\alpha2}$ cells (75 ± 13%—mean ± standard error of the mean) are confined to the ventrocaudal CA1 (Fig. 1a–f) and positive for somatostatin immunostaining (Fig. 1g, Supplementary movie 1). Injection of a Cre-dependent adeno-associated viral vector carrying channelrhodopsin 2 (ChR2) was used to express ChR2 in OLM$^{\alpha2}$ cells (Supplementary Fig. 1a, b). Light stimulation of OLM$^{\alpha2}$ cells in vCA1 but not dorsal CA1 (dCA1, Supplementary Fig. 1c–f) induced prominent theta oscillations (3–8 Hz) in mice under ketamine (Fig. 2a, b) or urethane (Fig. 2d–g) anesthesia. To prevent light-generated

artifacts at theta frequencies in LFP recordings, we used 16 Hz light stimulation[20,21]. Higher stimulation frequency and non-rhythmical stimulation also produced theta oscillations in the vCA1 (Supplementary Figs. 2 and 3). OLM$^{\alpha2}$ cell stimulation failed to elicit theta oscillations when PCs were inhibited, suggesting that theta induction depends on PC activity (Supplementary Fig. 4a–d). Distal dendrites of ventral PCs (the dendritic domain where OLM cells heavily synapse) abundantly express hyperpolarizing-activated current (Ih)[14]. Hence, we tested the role of this current in OLM$^{\alpha2}$ cell-induced theta. Microinjection of the Ih blocker ZD7288 (0.1 mM, 2 × 50 nL) at the *stratum lacunosum moleculare* (SLM) in urethane-anesthetized animals decreased the power of OLM$^{\alpha2}$ cell-induced theta (Supplementary Fig. 4e, f). Moreover, PCs phase-locked to the induced theta oscillations (Supplementary Fig. 5a, b), whereas the proportion of interneurons showing firing coupled to theta was not significantly different (Supplementary Fig. 5b), further supporting OLM$^{\alpha2}$ cell–PC interaction during theta. Current-source density analysis revealed the main source at the *stratum pyramidale* (SP) and a strongly attenuated source at the SLM, a pattern typical for theta2[22] (Fig. 2c). Notably, computer simulations of extracellular potentials from a PC receiving only rhythmical input from OLM cells also revealed a perisomatic source of theta oscillations (Supplementary Fig. 6). Systemic administration of atropine (Fig. 2d–g) or scopolamine (Supplementary Fig. 7) prevented OLM$^{\alpha2}$ cell-induced theta. In addition, optogenetically identified OLM$^{\alpha2}$ cell action potentials (AP) were strongly coupled to tail pinch-induced theta2 oscillations under anesthesia (Fig. 2h–k). In horizontal slices, where glutamatergic and GABAergic transmission were blocked, blue light did not induce rhythmicity in OLM$^{\alpha2}$ cells transfected with ChR2 under control conditions, while 16 Hz blue light stimulation during carbachol application led to rhythmical firing at theta2 frequency (Supplementary Fig. 8). By expressing the inhibitory proton pump Archaerhodopsin (Arch)[23] in the vHipp of Chrna2-Cre animals, we tested whether optogenetic inhibition of OLM$^{\alpha2}$ cells disrupts theta in urethane-anesthetized animals. Application of green light significantly decreased tail pinch-induced theta2 power (Fig. 3). To further explore the cholinergic nature of theta2 in urethane-anesthetized animals, we injected Arch in the MS/diagonal band of Broca (MS/DBB) of Chat-Cre animals (Supplementary Fig. 9a, b). Cholinergic inhibition strongly reduced tail pinch-induced theta power in the vCA1 (Supplementary Fig. 9c–e). Taken together, these data show that OLM$^{\alpha2}$ cell activation elicits vCA1 theta2 in anesthetized mice. Additionally, theta2 induction in the vCA1 requires PC activity and a basal cholinergic drive.

### OLM$^{\alpha2}$ stimulation induces theta2 in treadmill running mice.
We next tested if stimulation of OLM$^{\alpha2}$ cells could induce vHipp theta oscillations in awake animals. Mice were placed on a treadmill with preset speeds to avoid variations in theta1 frequency and amplitude. When running at 10 cm/s, an ~8 Hz (7.9 ± 0.6 Hz) theta1 peak appeared in the vCA1 (Fig. 4a, b). Optogenetic stimulation of OLM$^{\alpha2}$ cells in the vHipp did not affect this peak, but instead induced a prominent peak at ~7 Hz (6.8 ± 0.7 Hz) (Fig. 4a–c). The induced theta was atropine-sensitive (Fig. 4d–f), suggesting that stimulation of OLM$^{\alpha2}$ cells could generate theta2 also in behaving animals. Both theta1 and OLM$^{\alpha2}$ cell-induced theta2 peak frequency—but not amplitude—strongly correlated with the running speed (Supplementary Fig. 10a, b). To further investigate coexistence of theta1 and theta2, we simultaneously recorded LFPs in the CA1 of dHipp (Fig. 5a–c) and vHipp (Fig. 5d–f) during treadmill running. We found a theta2 peak next to the prominent theta1 peak in dHipp upon OLM$^{\alpha2}$ cell stimulation (Fig. 5b). Moreover, the presence of

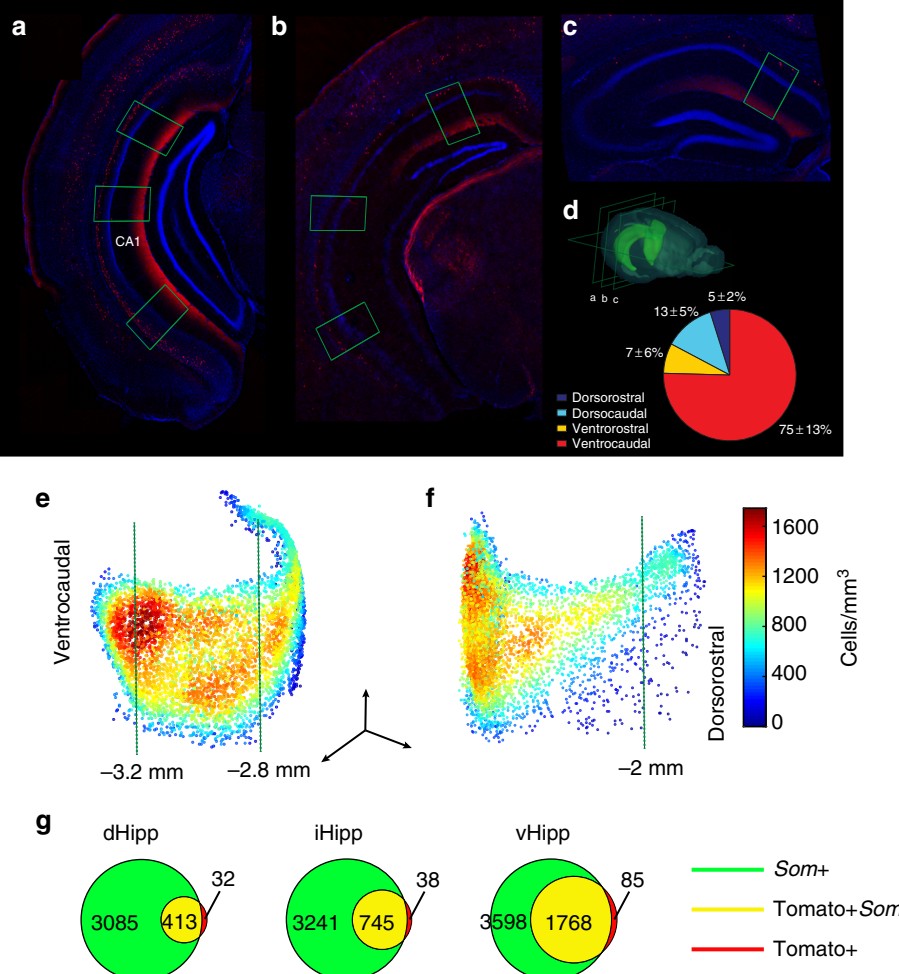

**Fig. 1** OLM$^{\alpha 2}$ cells display a gradient-like distribution along the dorsoventral axis. **a–c** Coronal sections displaying Chrna2+ cell expression in the ventrocaudal, intermediate, and dorsorostral hippocampus, respectively. Green rectangles represent the position of cell counting in the respective slices. **d** Chrna2+ cell quantification per hippocampal division ($n = 3$). **e, f** Averaged density plot ($n = 3$, data obtained using the CLARITY method, see Online Methods) showing the distribution of OLM$^{\alpha 2}$ cells from the caudal (**e**) and rostral (**f**) view (scale bars; x and z: 1 mm, y: 0.5 mm). Green punctuated lines represent relative distance from bregma. **g** Venn diagram illustrating the colocalization of somatostatin and tdTomato in Chrna2/tdTomato mice obtained using the CLARITY method (Supplementary Movie 1). The vast majority of Chrna2+ cells were Som+ along the dorsoventral axis

movement-elicited theta1 did not prevent theta2 induction, which had a higher amplitude in vHipp compared to dHipp.

Theta oscillations in the dHipp are known to modulate faster local oscillations[24]. Hence, we tested whether OLM$^{\alpha 2}$ cell-driven theta2 could modulate local gamma oscillations in vCA1. In mice running on a treadmill, theta1 phase in the *stratum radiatum* (SR) modulated gamma amplitude before light stimulation (Fig. 6a). During light stimulation, 60–80 Hz gamma oscillations were modulated by the induced theta2 (Supplementary Fig. 6a–c). In the presence of atropine, light stimulation failed to alter theta1–gamma coupling (Fig. 6d–f). Also, we did not observe any changes in theta–gamma coupling in dHipp upon OLM$^{\alpha 2}$ cell stimulation (Supplementary Fig. 11). These results imply that theta2-generating circuits modulate gamma-generating networks in the vCA1, and may, in similarity to theta1, play a role in the information flow between the entorhinal cortex, CA3 and CA1[25].

**OLM$^{\alpha 2}$ theta2 controls anxiety responses to predator odor.** Theta2 in rodents appears during arousal and vigilant conditions, such as innate anxiety in the presence of a predator or its smell[5].

Therefore, we tested whether theta2 induction by OLM$^{\alpha 2}$ cell stimulation would affect behavioral responses to predator smell. We placed cat hair (see Online Methods) in the center of a circular arena divided into three zones (Wall, Intermediate, and Center, Fig. 7a and Supplementary Fig. 12). OLM$^{\alpha 2}$ cells were light-stimulated at 16 Hz while animals explored the Intermediate and Center zones of the arena (Fig. 7b, c). We delivered light only when animals explored inner zones since vHipp theta activity was shown to appear specifically in the outer borders of a circular arena in an anxiety-related test[10]. During light stimulation, the frequency with which Chrna2-Cre/ChR2 mice crossed the border between the Wall and Intermediate zones was much higher than for control mice (OLM$^{\alpha 2}$ cells expressing only eYFP and stimulated with light—see Online Methods) (Fig. 7b, c). Most strikingly, Chrna2-Cre/ChR2 animals showed significantly higher crossing frequency from the Intermediate to Center zones and spent more time in the Center zone (Fig. 7c). Both groups exhibited theta1 (7.3 ± 0.4 Hz, average velocity 7 ± 1.2 cm/s) at the Wall zone (Fig. 7d, e). When exploring the Intermediate zone, where animals moved at a relatively faster pace (~25 cm/s), the frequency of theta1 increased to 9.5 ± 0.3 Hz (Fig. 6f). However, in this case, a prominent lower frequency theta2 (~8 Hz)

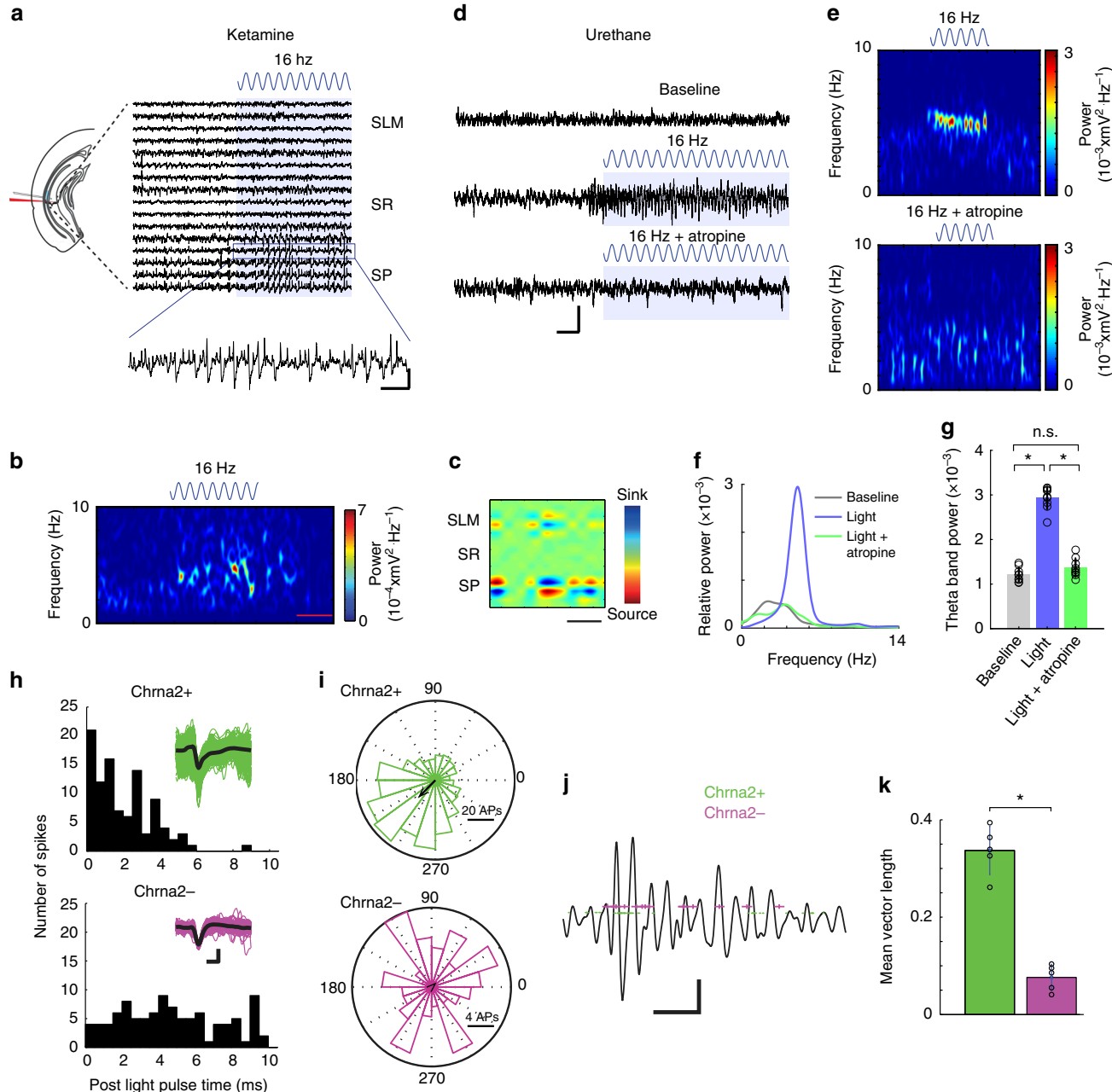

**Fig. 2** OLM$^{\alpha 2}$ cell stimulation generates atropine-sensitive theta oscillations in vHipp. **a** Representative raw LFP recordings from SP, SR, and SLM of vHipp CA1 strata before and during OLM$^{\alpha 2}$ cell stimulation by 16 Hz blue light in a ketamine-anesthetized mouse. The scale bar (*x*: 0.5 s and *y*: 0.05 mV) refers to the scale of the zoomed period. **b** A representative spectrogram 20 s before, during, and after light stimulation. Scale bar: 10 s. **c** An example of current-source density during OLM$^{\alpha 2}$ cell stimulation. Note the weaker dipole in SLM compared to SP. Scale bar: 150 ms. **d** Representative recording traces from SP before and during OLM$^{\alpha 2}$ cell stimulation under urethane anesthesia. Note that atropine prevents induction of theta activity. Scale bar, *x*: 1 s and *y*: 0.1 mV. **e**, **f** Spectrograms (**e**) and power spectra (scale bar: 10 s) (**f**) for the recordings shown in **d**. **g** Group statistics (*n* = 9, \**p* < 0.0001, one-way repeated-measures ANOVA, Bonferroni post-hoc test). **h** Examples showing units recorded with "optotrodes" in the SO that respond (Chrna2+, green) or not (Chrna2−, magenta) to the light stimulation (waveforms are shown in the insets—scale bars: 0.4 ms/0.05 mV) under urethane anesthesia (theta elicited by tail pinch). **i** Polar histograms showing Chrna2+ and Chrna2− unit firing relationship to theta phase. **j** LFP segment overlaid with firing times of Chrna2+ and Chrna2− units. Scale bars: 0.5 s/0.05 mV. **k** Mean spike-field coupling strength for firing times Chrna2+ and Chrna2− units (*n* = 10 units/3 mice, \**p* = 0.000006, *t*-test). Error bars in all bar graphs represent SEM

also appeared in both animal groups, but with significantly larger amplitude in light-stimulated Chrna2-Cre/ChR2 animals (Fig. 7f, g). Note that we compare Chrna2-Cre/ChR2 to control animals solely in the Wall and Intermediate zones, since control animals rarely visited the Center zone (Fig. 7c). In Chrna2-Cre/ChR2 animals, the amplitude of the induced theta2 peak (6–8 Hz) correlated with exploration in the Center zone, while theta1

(8–10 Hz) was associated with movement, independent of the zone (Fig. 7h–j). Theta2 in the Intermediate and Center zones also modulated 60–80 Hz gamma in the SR of vCA1 (Supplementary Fig. 13). Activation of OLM$^{\alpha 2}$ cells did not alter smell perception for aversive or non-aversive scents (Supplementary Fig. 14), suggesting that the observed effects were not mediated by an inability to discriminate the smell stimulus.

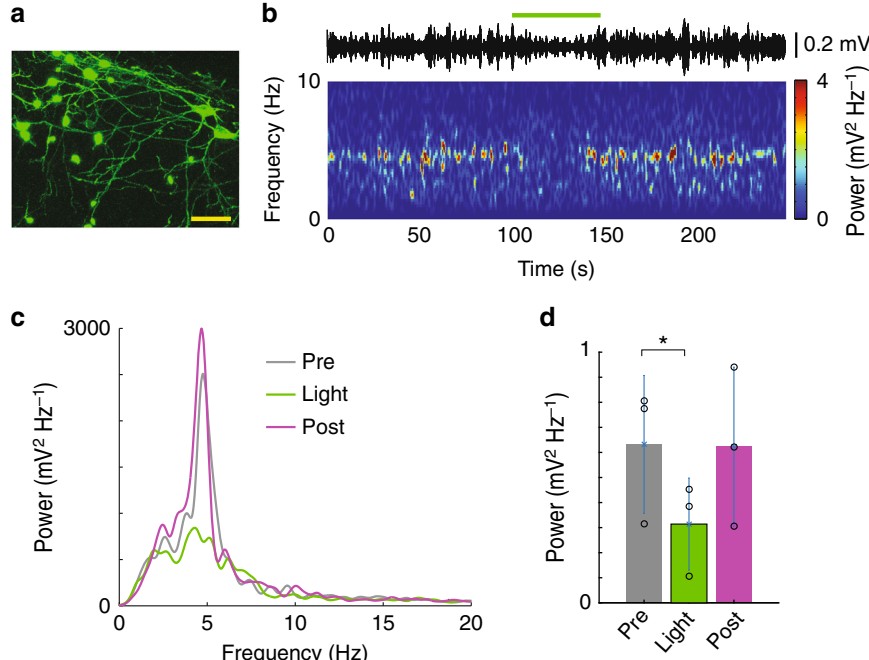

**Fig. 3** OLM$^{\alpha 2}$ cell inhibition decreases theta activity in urethane-anesthetized animals. **a** Confocal microscopy image of Arch expression in OLM$^{\alpha 2}$ cells (reported by EGFP green fluorescence) (scale bar: 20 μm). **b** Spectrogram following a tail pinch (starting at time 0) showing the effect of green light in an animal expressing Arch in OLM$^{\alpha 2}$ cells. **c** Example of a power spectrum density plot showing power before (black trace), during (magenta trace), and after (green trace) light delivery. **d** Mean theta power before (Pre), during (Light), and after (Post) light delivery ($n = 3$ mice, $^*p = 0.03$, paired $t$-test). Error bars in all bar graphs represent SEM

We next sought to investigate the effect of OLM$^{\alpha 2}$ cell inhibition on vHipp oscillatory activity in the same predator odor behavioral task. To this end, we injected Chrna2-Cre animals with the inhibitory opsin Archaerhodopsin (Chrna-Cre/Arch). We implemented a feedback system that detected theta2 in real-time to control a green laser to optogenetically inhibit OLM$^{\alpha 2}$ cells when animals were moving (Fig. 8a, Supplementary Fig. 15a). Without OLM$^{\alpha 2}$ cell activation, theta2 was mostly detected outside the Intermediate region, in the zone named Extended Intermediate (Supplementary Fig. 15a). OLM$^{\alpha 2}$ cell inhibition resulted in Chrna-Cre/Arch mice spending more time in the Wall zone and in a decreased crossing frequency from Wall to Extended Intermediate and from Extended Intermediate to Intermediate zone (Fig. 8b, c). Furthermore, the time spent in the Extended Intermediate and Intermediate zones was significantly reduced in the light-inhibited Chrna2-Cre/Arch animals (Fig. 8d). In the Wall zone, both control and Chrna2-Cre/Arch groups displayed single theta1 peaks (Fig. 8e, f). In the Extended Intermediate zone, control animals displayed both theta1 and theta2 peaks, whereas optogenetically inhibited Chrna2/Cre-Arch animals showed significantly reduced power of theta2 (Fig. 8g, h). Average speed in control and Chrna2-Cre/Arch animals did not differ significantly in the Extended Intermediate and Intermediate zones (Supplementary Fig. 15b). To further dissociate the effect of animals' movement speed on OLM$^{\alpha 2}$ cell-driven theta activity, we correlated running speed and frequency using the method described in ref. [26] (Fig. 9). Both theta1 and theta2 frequency increased with running speed; however, theta2 only appeared in the vicinity of the predator odor, in the Intermediate zone (Fig. 9a–c). This data indicates that OLM$^{\alpha 2}$ cell-driven theta2 appearance is related to the state of the animal anxiety, rather than movement-related activity. Furthermore, we did not observe any effect of OLM$^{\alpha 2}$ cell inhibition in other oscillation frequencies (Supplementary Fig. 16a, b). These results show that OLM$^{\alpha 2}$ cell inhibition in Chrna2-Cre/Arch animals abolishes the

appearance of theta2, without affecting either theta1 power, theta1/running speed relationship, or other oscillation frequencies.

Theta2 has been classically associated with immobility[3–5]. Thus, we further sought to understand the effect of OLM$^{\alpha 2}$ cell inhibition on the oscillatory activity and behavior in immobile animals. Similarly to the moving animals, we inhibited OLM$^{\alpha 2}$ cells in a closed loop experiment, with the difference that green light was delivered when the criteria of immobility and theta2 appearance were fulfilled (see Methods). The theta2 detected in immobile animals was strongly reduced in the light inhibited Chrna2-Cre/Arch group when compared to control mice (Fig. 10a–d). In the Extended Intermediate and Intermediate zones, where theta2 was detected, both control and Chrna2-Cre/Arch animals were predominantly moving (Fig. 10e). Taken together, these results show that optogenetic activation of OLM$^{\alpha 2}$ cells induces theta2 and decreases anxiety, whereas OLM$^{\alpha 2}$ cell inhibition decreases theta2 power and increases anxiety responses to predator smell in both moving and immobile animals.

**Discussion**

Here, we explored a specific interneuron population, OLM$^{\alpha 2}$ cells, in the vHipp and found it to be both sufficient and necessary to maintain a specific type of theta oscillation and to control behavior. Using electrophysiology and optogenetics, we show that OLM$^{\alpha 2}$ cell stimulation in vivo generates cholinergic-dependent theta2 oscillations in both anesthetized and freely behaving mice. We further show that theta2 can coexist with movement-driven theta1. Furthermore, theta2 induced by OLM$^{\alpha 2}$ cell stimulation was directly related to a considerable increase in risk-taking behavior, whereas OLM$^{\alpha 2}$ cell inhibition significantly decreased naturally occurring theta2 and risk-taking behavior in a predatory odor innate anxiety test. While inhibitory interneurons have gained center stage in rhythmogenesis, with several studies

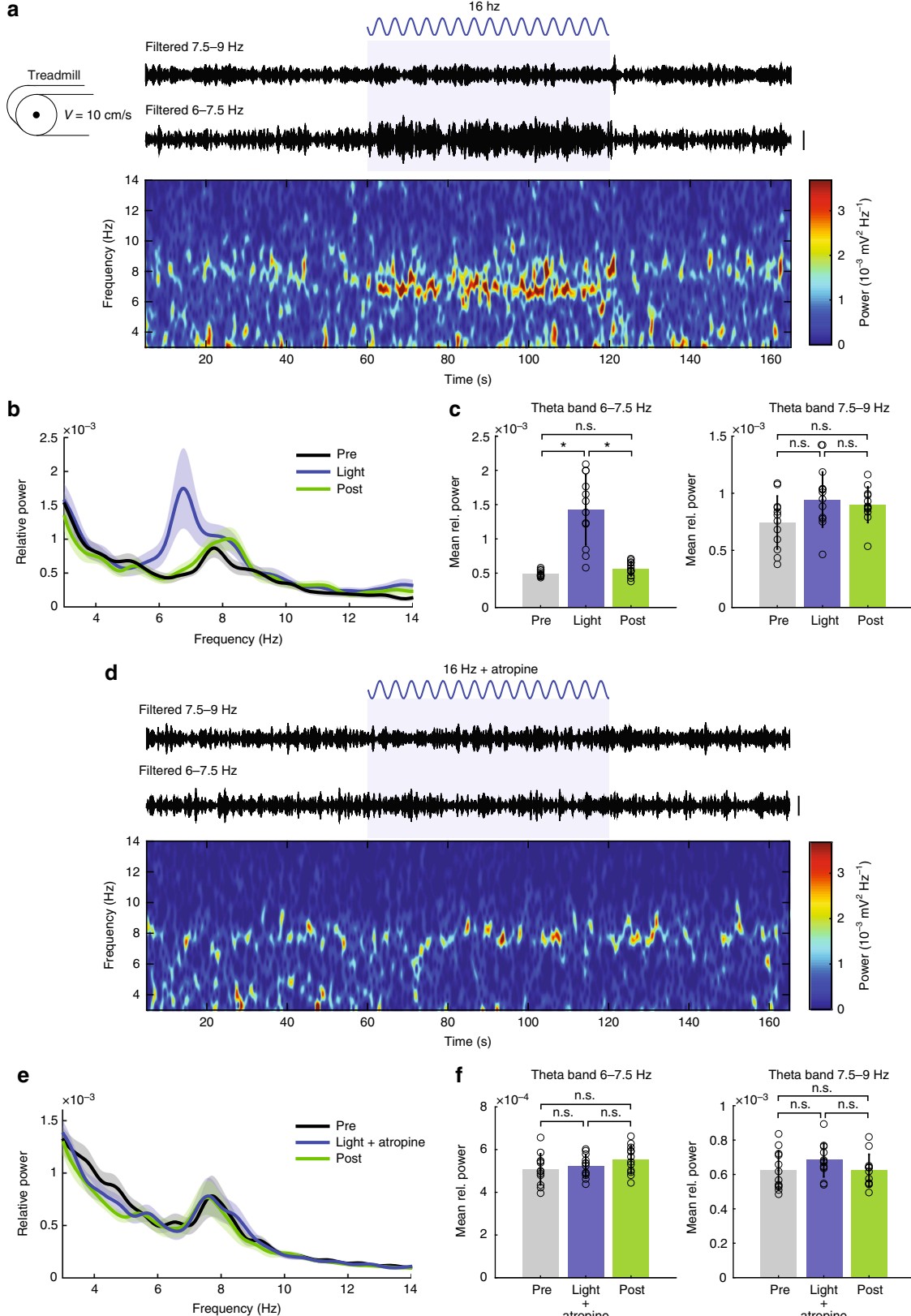

**Fig. 4** OLM$^{\alpha 2}$ cell stimulation generates type 2 theta in vHipp in running animals. **a** Filtered traces and spectrogram showing increase in theta2 (~7 Hz) activity during OLM$^{\alpha 2}$ cell stimulation, without affecting movement-related theta1 (7.5–9 Hz). **b**, **c** Group statistics ($n = 12$, *$p < 0.0001$, n.s. = not significant, repeated-measures ANOVA). **d–f** Atropine application precludes theta2 induction by OLM$^{\alpha 2}$ cell stimulation, leaving theta1 unaffected ($n = 8$, n.s. = not significant, repeated-measures ANOVA). **a** and **d** scale bars: 1 mV. Error bars in all bar graphs represent SEM

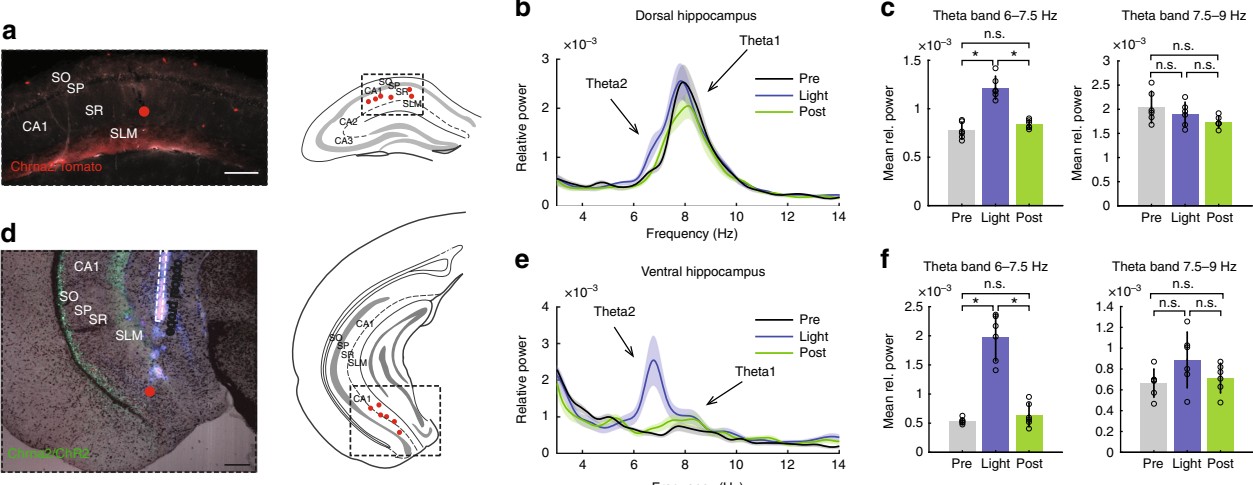

**Fig. 5** Theta2 peak induced in vHipp affects dHipp. **a** Detected probe positions in dHipp. Scale bar: 0.2 mm. **b** Averaged power spectra of recordings in dHipp 60 s before, during and after OLM$^{\alpha 2}$ cell stimulation. Note prominent movement-related theta peak at ~8 Hz while animals were running on a treadmill at 10 cm/s velocity, while during light stimulation a less visible theta2 peak at ~7 Hz appears. **c** Bar plots of mean theta power showing a significant increase in 6–7.5 Hz theta peak ($n = 6$, *$p < 0.001$, repeated-measures ANOVA), while no difference in 7.5–9 Hz was detected. **d** Detected probe positions in vHipp and, in a few cases, ventral subiculum. Since we observed similar effects in both vHipp and ventral subiculum (both contain OLM$^{\alpha 2}$ cells), we have pulled the data together. Scale bar: 0.2 mm. **e** Averaged power spectra of ventral recordings. Note smaller movement-related ~8 Hz peak compared to dHipp[8,9], while prominent theta2 peak ~7 Hz was induced by OLM$^{\alpha 2}$ cell stimulation. **f** Bar plots of mean theta power for 6–7.5 and 7.5–9 Hz theta band ($n = 6$, *$p < 0.0001$, repeated-measures ANOVA), showing a significant increase in the theta2 (6–7.5 Hz), but not theta1 (7.5–9 Hz). Error bars in all bar graphs represent SEM

pointing to the role of specific interneuron types in relation to type 1 theta in dHipp[7,27,28], little is known about the cellular substrates underlying type 2 theta. To our knowledge, this is the first demonstration of a causal role of a specific interneuron population in vHipp in generating theta2 oscillations and controlling behavior.

Several studies support the view that different classes of interneurons preferentially fire in specific phases of theta or gamma oscillations, and thus may be causally involved in producing them[28–32]. Experiments using optogenetics have selectively stimulated neurons expressing genetic markers such as parvalbumin, and found that these neurons can drive oscillations at the gamma frequency[29,33]. However, parvalbumin expression is present in many different interneuron subtypes[34], as it is also the case with somatostatin[19,32], another commonly used marker. In contrast, the Chrna2/Cre line is one of the most specific markers for a given hippocampal interneuron subtype[35]. The occurrence of optoelectric artifacts in these previous studies is also a concern, since these studies often use a stimulation frequency that is the exact same as the one observed in LFP recordings as a stimulated peak, making it impossible to separate a physiological response from artifacts (see ref. [21] for a discussion). We have addressed this problem by using a frequency of 16 Hz, which is outside the range of the LFP band we analyzed, thus increasing the reliability of our observations.

It has been suggested that theta2 in CA1 is regulated by CA3 and MS inputs while theta1 is controlled by both the entorhinal cortex and the MS[36,37]. Lesions of the entorhinal cortex strongly attenuate theta1 while theta2 is unaffected[37]. These results are in line with our CSD analysis demonstrating that OLM$^{\alpha 2}$ cell-induced theta2 has a strong perisomatic source in SP, but a weak source at SLM. A possible mechanism for the perisomatic source of theta2 is that OLM$^{\alpha 2}$ cells could inhibit back-propagating AP in active distal dendrites. These results are further supported by our previous study demonstrating that OLM$^{\alpha 2}$ cell activity facilitates CA3 input while inhibiting a direct input from the entorhinal cortex[18]. Another possible

mechanism for a perisomatic theta2 is that intrinsic excitation of PC distal dendrites by voltage-gated currents could compensate for the lack of entorhinal cortex excitatory drive. In support, OLM$^{\alpha 2}$ cells synapse in distal PC dendrites in the ventral CA1 that, differently from dorsal PC, show a strong expression of hyperpolarization-activated cyclic nucleotide-gated (HCN) channels[14]. Further, we found that blocking hyperpolarizing-activated current (Ih) in the SLM prevented the generation of theta2 by OLM$^{\alpha 2}$ cell stimulation. Hence, strong and rhythmical inhibition from OLM$^{\alpha 2}$ cells targeting HCN-rich dendritic compartments could produce rhythmical rebound depolarization. This, in turn, could compensate for theta2 generation in the absence of entorhinal inputs.

We found that OLM$^{\alpha 2}$ cell-driven theta originates in the vHipp and extends to the dHipp (Fig. 5). A recent study reported that local cholinergic stimulation in MS enhances type 2 theta in the dHipp of urethane-anesthetized animals[6]. It is possible that this activity may originate from the vHipp, as suggested by our data. This would be coherent with the significantly denser cholinergic innervation from the MS in the vHipp than in the dHipp (http://mouse.brain-map.org/). The coexistence of theta1 and theta2 in freely moving animals may have been difficult to observe in previous dHipp studies due to the prominent theta1 in this region. Note that a single theta source in the dHipp that reaches the vHipp through "traveling waves"[8,9] would not account for the lower theta frequency found in vHipp in certain emotion-related behaviors[10,11]. We also found that OLM$^{\alpha 2}$ cell-driven theta can modulate gamma oscillations in vHipp, even in the presence of theta1 (Fig. 6) without affecting theta–gamma comodulation in dHipp. Hence, the circuits responsible for the emergence of emotion-related theta2 in vHipp may be distinct from circuits generating dorsal theta1.

Interestingly, theta2 frequency was correlated with speed. In contrast, we found no theta2 power/running speed correlation, similar to a previous study[9]. Some studies have suggested that the relationship between theta1 frequency and running speed is associated with changes in cholinergic drive to hippocampus/

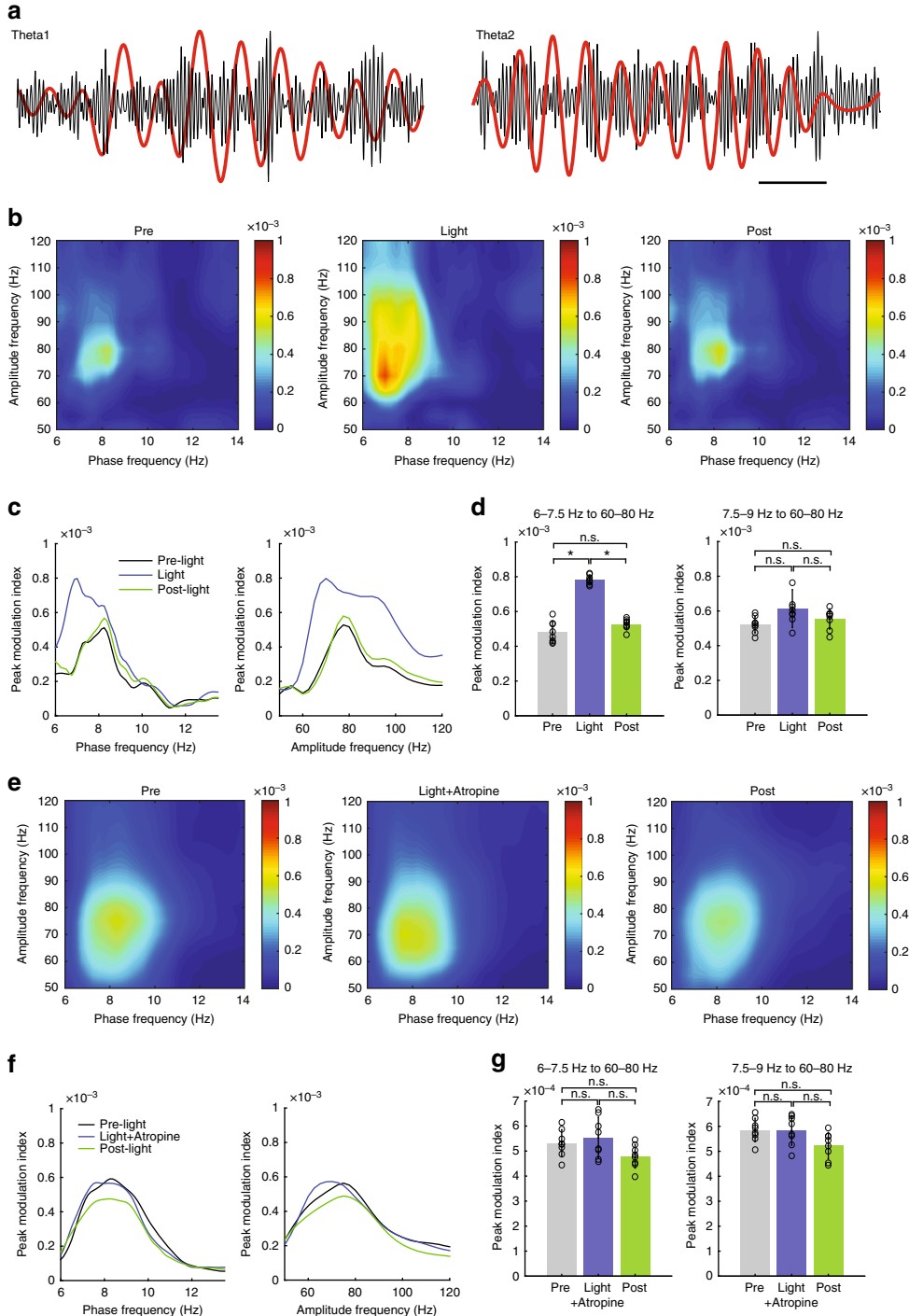

**Fig. 6** Theta2 modulates 60–80 Hz gamma activity in the vHipp stratum radiatum. **a** Examples of LFP traces (normalized by their maxima) filtered at theta1 (before light) and theta2 (during light) and gamma (bandpass 50–100 Hz). Scale bar: 250 ms. **b** Representative comodulograms showing theta–gamma phase–amplitude coupling measured before, during, and after OLM$^{\alpha 2}$ cell stimulation while animals were running on a treadmill at 10 cm/s. No difference in gamma band power before and during light stimulation was found (data not shown). **c** Peak comodulation frequencies for theta and gamma oscillations. 7.5–9 Hz (theta1) couples to gamma before and after light stimulation. The arrow points to the modulation shifts to ~7 Hz during light (equivalent to the power peak induced by OLM$^{\alpha 2}$ cell stimulation shown in Fig. 4b), while modulated gamma increases especially for ~70 Hz. **d** Group statistics show that theta modulation of gamma increased for 6–7.5 Hz (theta2) ($n = 8$, *$p < 0.0001$, repeated-measures ANOVA), whereas no difference for theta1 was observed ($n = 8$, n.s. = not significant). **e–g** Atropine prevented theta2–gamma comodulation, leaving theta1–gamma comodulation unaffected. Error bars in all bar graphs represent SEM

entorhinal cortex[38]. Ventral OLM$^{\alpha 2}$ cells are strongly modulated by cholinergic inputs[39], and changes in cholinergic drive might also modulate theta2 frequency. Acetylcholine is known to increase resonance frequency through modulation of Ih and m-current both in vivo and in vitro[40,41]. It is thus plausible that cholinergic inputs link running speed with theta2 frequency.

Dorsal and ventral OLM cells might comprise two different cell populations. One study has shown that two OLM cell populations differentially contribute to network activity depending on their

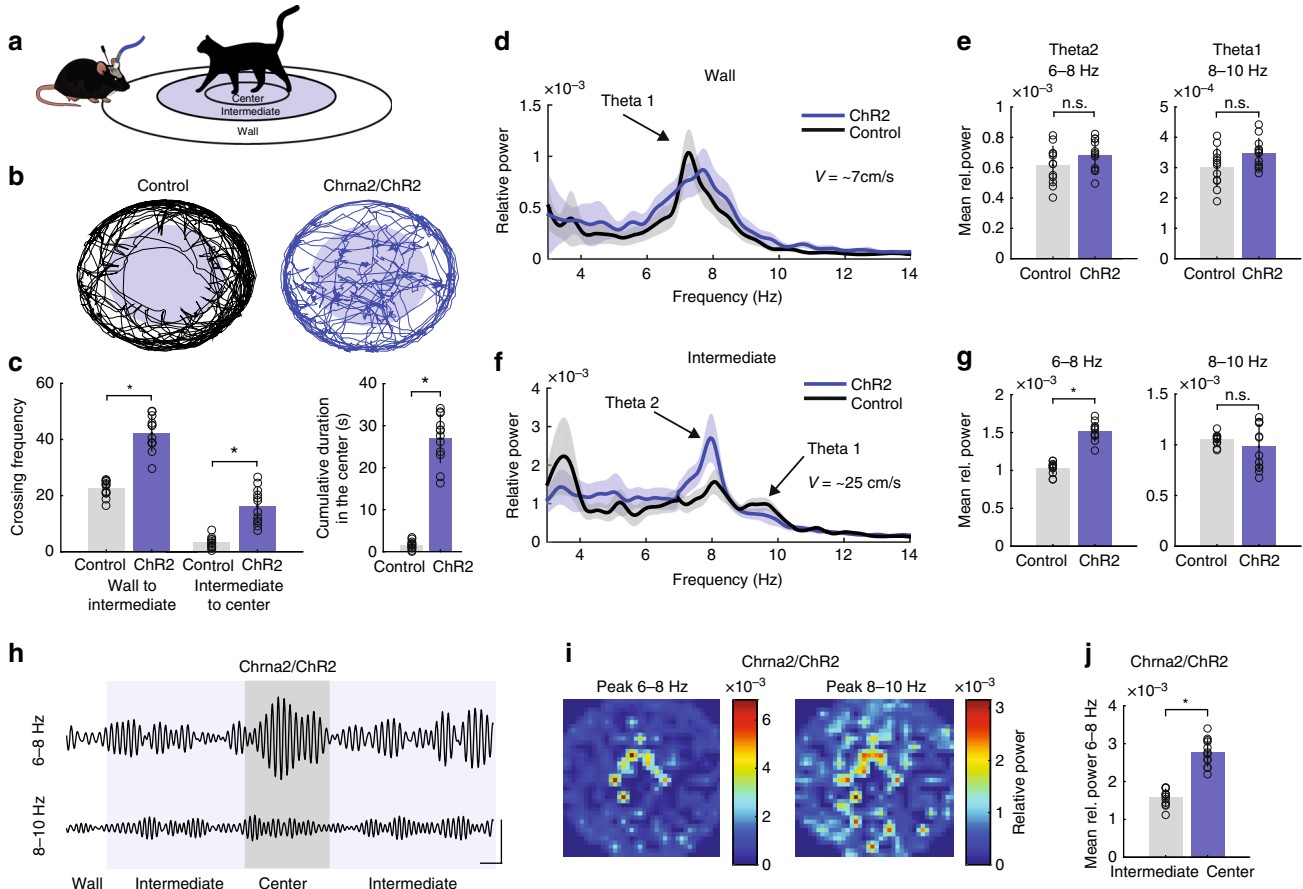

**Fig. 7** OLM$^{\alpha2}$-induced theta decreases anxiety responses to predator odor. **a** Experimental setup. Cat hair was placed in the Center zone of the circular arena. Intermediate and Center (in blue) were light-stimulated zones. The cat and mouse cartoon were produced by the authors using the free computer programs GIMP and Inkscape. **b** Tracking paths of one Control (Chrna2-Cre animal injected with eYFP virus) and one Chrna2/ChR2 animal. Both groups were stimulated with light (16 Hz sinusoid). **c** (left) Significant difference in crossing frequency between Control and Chrna2/ChR2 animals from Wall to Intermediate ($n = 12$ per group, $*p < 0.0001$, $t$-test) and Intermediate to Center ($n = 12$ per group, $*p < 0.0001$, $t$-test). (right) Chrna2/ChR2 animals spend significantly longer exploring the Center ($n = 12$ per group, $*p < 0.0001$, $t$-test). **d**, **e** In the Wall zone, only movement-related theta1 appears, showing no difference between the groups. **f**, **g** In the Intermediate zone, next to the theta1, a lower frequency type 2 theta appeared in both groups, with higher amplitude in Chrna2/ChR2 animals ($n = 12$ per group, $*p < 0.0001$, repeated-measures ANOVA). **h–j**. Significant increase in theta2 power during exploration of Center versus Intermediate zones (scale bars: 0.5 s/0.5 mV). Representative heat maps show increase of theta2 (6–8 Hz) towards the Center of the arena, whereas movement-related theta1 appears in all zones. Error bars in all bar graphs represent SEM

embryonic lineage[42]. A recent modeling paper[43] proposed that, depending on the presence of HCN channels on their dendrites, OLM cells could be recruited by either high or low theta frequency. These results concur with the experimental data from our laboratory showing a gradient of Ih in OLM cells, difference in membrane resonance properties (unpublished) and responsiveness to cholinergic agonists along the dorsoventral axis[44]. We have previously shown that OLM$^{\alpha2}$ cells receive cholinergic input from the MS[18]. Furthermore, using the same Cre line as used here, a recent study has shown that septal cholinergic inputs to the ventral CA1 OLM$^{\alpha2}$ cells control hippocampal output to the entorhinal cortex[39]. Another study suggests that glutamatergic input from the MS acts on putative OLM cells in the dHipp, controls locomotion speed and contributes to the generation of movement-related type 1 theta by facilitating both CA3 and entorhinal cortex inputs to the CA1 hippocampal region[7]. There is also evidence that the MS GABAergic axons in dHipp are involved in the generation of theta oscillations[45,46]. Thus, dorsal and ventral OLM cells (and other hippocampal interneurons) are likely to receive distinct MS/DBB innervation and could be differently involved in multiple types of theta activity.

The "sensorimotor integration model" proposed by Bland and colleagues[3,47] suggests that theta2 is relevant for the initiation and maintenance of movement. It also suggests that theta2 can be generated in isolation during immobility, but theta1 and theta2 occur coincidentally when the theta1 subsystem is active[48]. Our observations support this view as we experimentally show that theta1 and theta2 can coexist. However, we did not observe spontaneous theta2 in animals unless they were exposed to an anxiety-related test (e.g., in previously trained animals running on a treadmill). Our results indicate that theta2, in addition to representing a movement-related signal, strongly relates to the state of anxiety.

Theta2 has been previously associated with anxiogenic stimuli[4], but here we show that theta2 increases with risk-taking behavior associated with predator smell. It has been shown that the amplitude of theta activity in the vHipp is stronger when animals are in the safe zone (borders) in an anxiogenic context (open field)[10]. Another study[49] reported that chemogenetic activation of MS cholinergic neurons slows down theta frequency in the medial entorhinal cortex without changing locomotor speed, in agreement with our results. In an open field test, the authors showed

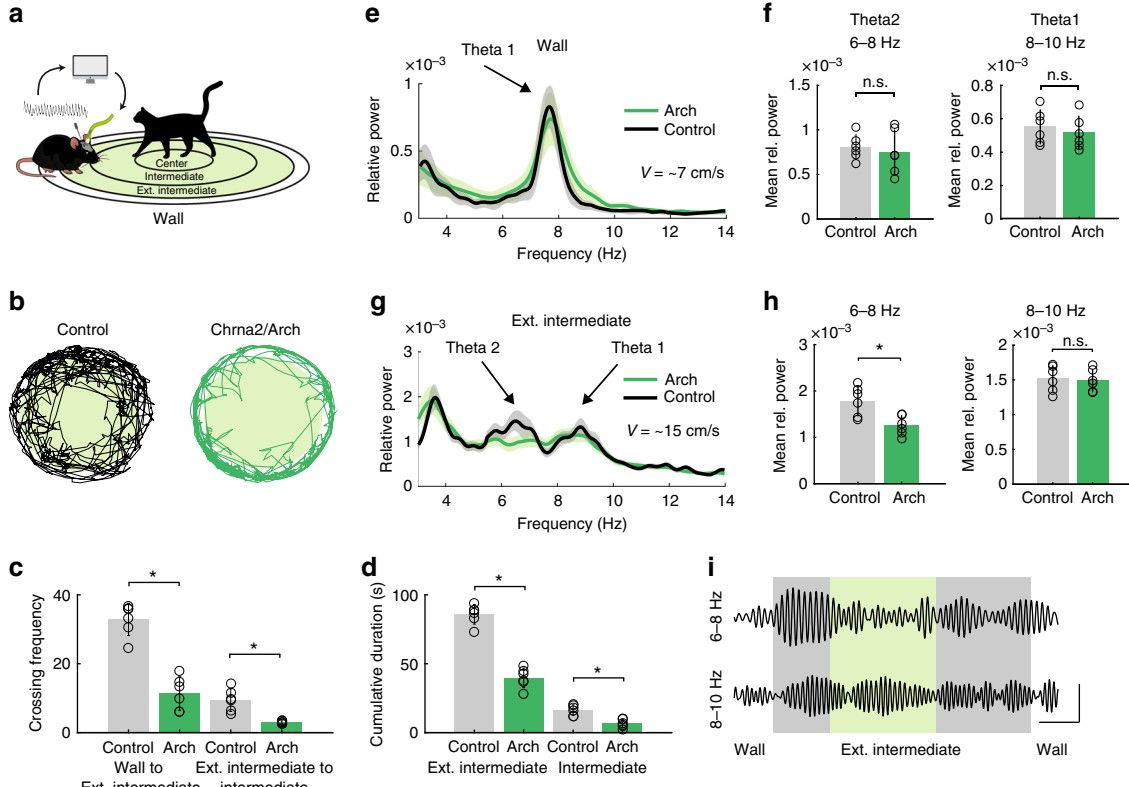

**Fig. 8** OLM$^{\alpha2}$ cell inhibition decreases the power of naturally occurring theta2. **a** Experimental setup. Naturally occurring theta2 was inhibited by green light in a closed-loop manner in the regions shaded in green. Cat hair was placed in the Center zone. **b** Tracking paths of one Control (Chrna2-Cre animal injected with eYFP virus) and one Chrna2/Arch animal with light delivery. **c** Crossing frequency in control and Chrna2/Arch animals from Wall to Ext. Intermediate ($n = 6$ per group, *$p < 0.0001$, $t$-test) and Ext. Intermediate to Intermediate ($n = 6$ per group, *$p < 0.0001$, $t$-test). **d** Chrna2/Arch animals spend significantly less time exploring the Ext. Intermediate and Intermediate zones ($n = 6$ per group, *$p < 0.0001$, $t$-test). **e**, **f** In the Wall zone, only movement-related theta1 is observed. **g**, **h** In the Ext. Intermediate zone, light-inhibited Chrna2/Arch animals displayed reduced theta2, but not theta1 power ($n = 6$ per group, *$p < 0.0001$, $t$-test). Note that we here normalized for movement by exclusively considering LFP epochs when animals were continuously walking ($\geq 2$ s). **i** Representative traces showing the effect of light-induced inhibition on theta2 and theta1 amplitude (scale bars: 1 s/0.1 mV). Error bars in all bar graphs represent SEM

that MS cholinergic activation results in decreased time spent in the center of the arena, indicating increased anxiety. Thus, we believe that the generation of theta2 by OLM$^{\alpha2}$ cell stimulation could be related to the perception of safety. Another possible explanation for the increase in risk-taking behavior is that theta2 could be associated with the arousal state of an animal. A recent study[50] has shown that a fearful stimulus leads to an increase in the low-frequency theta rhythm, whereas a social stimulus leads to an increase in the high-frequency theta rhythm. In addition, different brain regions were synchronized depending on the behavioral paradigm as well as the slow or fast theta appearance[50]. The authors suggested that the low-frequency theta, possibly theta2, might be associated with information processing during a "negative" arousal state, while high-frequency theta1 might be linked to a "positive" arousal state, such as during exploration or voluntary movements[50]. However, theta2 is also elicited in male animals in the presence of female subjects[5] indicating that theta2 appearance might not be unique to anxiety. The emergence of OLM$^{\alpha2}$ cell-driven theta2 could trigger an increase in arousal associated with an increase in risk-taking behavior and lead to different behavioral outputs depending on the behavioral context in which the animal is involved. It will be interesting to investigate the relationship between theta1 and theta2 in different behavioral paradigms. Moreover, given the high connectivity between vHipp, prefrontal cortex, and amygdala[51], it is likely that OLM$^{\alpha2}$ cell-induced theta2 has a major role

in synchronizing the vHipp with these regions. Finally, the possibility of controlling theta2 by modulating OLM$^{\alpha2}$ cells opens the door for future studies that aim to understand how dHipp theta1 and vHipp theta2 might coordinate cognitive and emotional processes.

## Methods

**Subjects**. Adult (2–6 months old) Chrna2-Cre, *Gt(ROSA)26Sor$^{tm14(CAG-tdTomato)}$ $^{Hze}$/J* (R26$^{tom}$, Jax Stock 007909), and Chat-Cre males (Jax Stock 006410) were used. Animals were housed in a group (up to 5 animals/cage), kept in a 12-h light on/light off cycle (7 a.m.–7 p.m.), and maintained at $21 \pm 2$ °C. All animal procedures were approved by the local Swedish and Brazilian ethical committees (C3/12, C132/13, C135/14, C45/16, Uppsala Animal Ethics Committee, Jordbruksverket and 052/2015, Animal Ethics Committee (CEUA) of the Federal University of Rio Grande do Norte).

**Virus injection**. Chrna2-Cre mice were anesthetized with 2% isoflurane. Animals were placed in a stereotaxic frame (Stoelting) and injected with the adeno-associated virus vector AAV2.EF1a.DIO.hChR2(H134R)-EYFP.WPRE.hGH, AAV2/EF1a—DIO-eArch3.0-eYFP, AAV2CaMKIIa-eNpH3.0-EYFP or control viruses (University of North Carolina Vector Core Facility) at a titer of $1 \times 10^{12}$ particles/ml. We used the EYFP version in all vectors since vectors that contained the red light-emitting fluorophore mCherry did not produce adequate expression. Vectors (0.5 μl) were injected unilaterally at three consecutive depths in the hippocampus (AP: −3.2 mm, ML: −3.8 mm, and DV: 2.5/3.0/3.6 mm) for a total volume of 1.5 μl. For injections in the MS of Chat-Cre animals, we used following coordinates: AP+ 0.8 mm, ML+ 0.7 mm, 10° angle insertion or AP+ 0.7–0.9 mm, midline insertion at 0° angle. Total volume of 1 μl virus (AAV2/EF1a—DIO-eArch3.0-eYFP) was injected at three different depths, between 2.8 and 4 mm

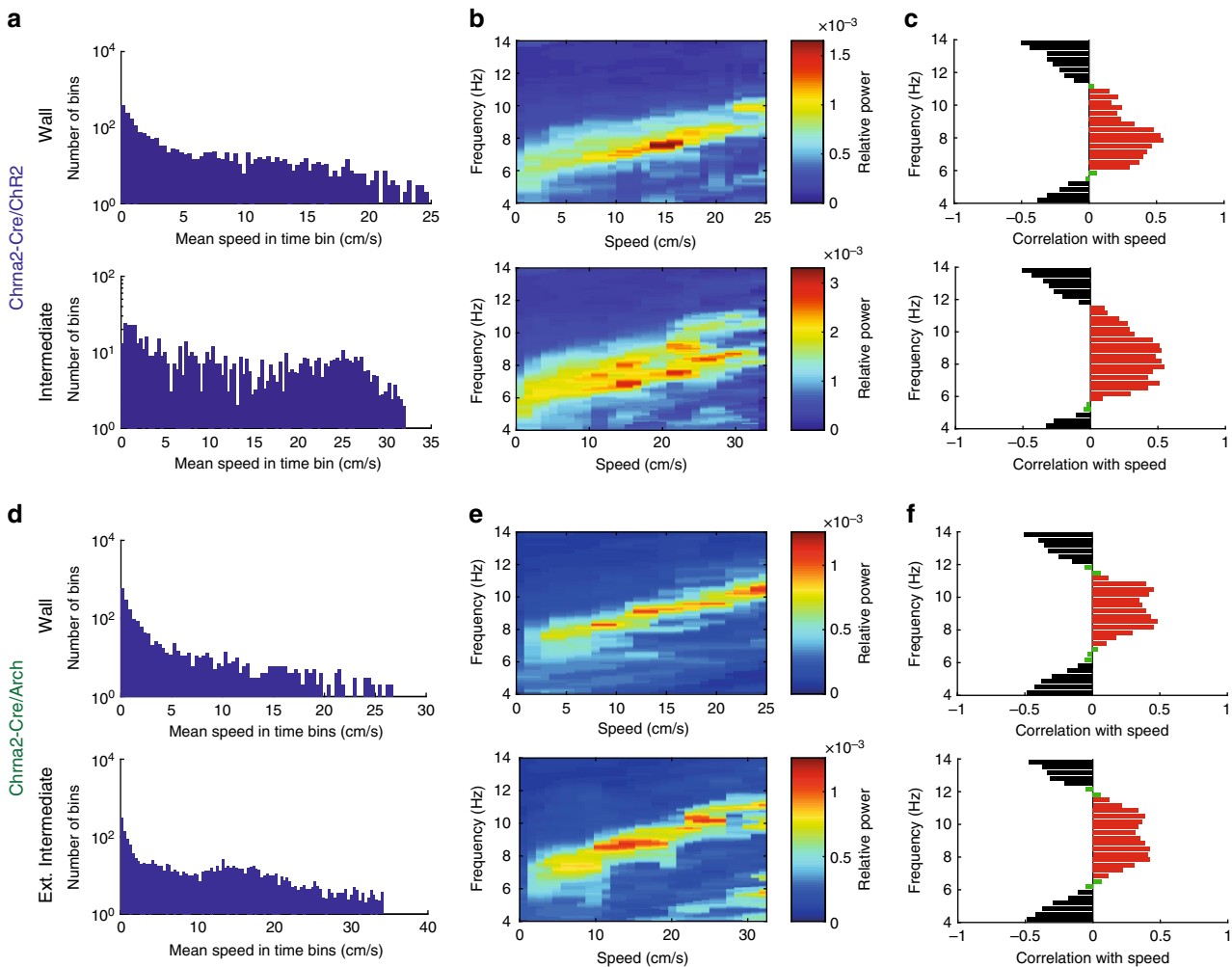

**Fig. 9** Theta2 frequency, similarly to theta1 frequency, is speed-dependent. In the predator odor test, a slower theta2 frequency appears in the vicinity of the odor cue in Chrna2/ChR2 animals. OLM$^{\alpha 2}$ cell inhibition precludes theta2 appearance without affecting theta1. **a** One session (10 min) was divided into 200 ms time bins. The histogram shows the distribution of mean running speeds separately for time bins when the animal was in the Wall or Intermediate zone. Notice higher speeds in the Intermediate zone and that there are fewer time bins in this zone because the animal spends more time in the Wall zone. **b** The running speed distribution was further binned into 30 speed bins. The relative power for each frequency was computed for all speed bins using the discrete Fourier transform. Note the existence of two theta frequencies in the Intermediate zone, both correlated with speed. **c** Significant positive correlations between power and speed (red bars, $p < 0.001$, two-way ANOVA) occurred for frequencies between 6 and 11 Hz. Significant negative correlations (black bars, $p < 0.001$, two-way ANOVA) were seen in the frequencies under 6 Hz and above 11 Hz. The analysis was proposed by Ahmed and Metha[26]. **d–f** The same analysis as in **a–c**, but for Chrna-Cre/Arch animals. Note that OLM$^{\alpha 2}$ cell inhibition did not affect theta1 and its correlation with running speed, but abolished theta2

(10° angle insertion), or at 3.0–3.4 mm (midline, 0° insertion). The flow rate was 200 nl/min; after each infusion, the needle was left in place for 1 min. The scalp incision was sutured and animals were housed in a P2 facility after the injections. We confirmed expression by post-hoc histological analysis of hippocampal sections.

**In vivo electrophysiology and signal processing**. Extracellular recordings in anesthetized animals were performed under ketamine (70 mg/kg)/midazolam (20 mg/kg) or urethane (21% solution; 1.2–2.0 g/kg) anesthesia. Note that under this dose of anesthesia, we observe no or little spontaneous theta activity (Fig. 1), allowing us to reliably analyze induction of theta activity by optogenetic activation of OLM$^{\alpha 2}$ cells. Acute silicon-substrate multi-channel A1–16-electrode probes (16 recording sites spaced 50 or 100 μm apart and distributed along a single shank, Neuronexus) were inserted in CA1 of the right vHipp[16] using a stereotaxic frame (AP: −3.0 mm, ML: −3.5 mm, and DV: 3.6 mm). Through a second orifice, an optical fiber (Thorlabs, 200-μm diameter, 0.39 numerical aperture) was inserted (AP: −3.2 mm, ML: −3.7 mm, and DV: 2.8 mm) in a 10–30° angle in relation to the recording probe. For current source density (CSD) analysis in vHipp of anesthetized animals (Fig. 1c), we have placed recording electrodes horizontally to record from all strata (AP: −3.2 mm, ML: −4.6 mm, and DV: 1.5 mm). In some experiments, we have used "optotrodes"[21] consisted of 8 × 25 μm insulated

tungsten wires glued to a 200 μm fiber optics. For infusion of ZD7288 a 200 μm canula metal canula was placed adjacent to SLM (AP: −3.0 mm, ML: −3.2 mm, and DV: 3.6 mm). OLM units were recorded with modified optotrodes (twisted 4 × 25 μm insulated tungsten wires glued to a 200 μm fiber optics)[52].

A blue 473 nm solid state laser (analog modulation) was driven by a sinusoid-like function (0 to maximum amplitude at various frequencies) generated by a DAQ card and a custom Matlab program. Laser power at the tip of the fiber was around 100 mW/mm² (Shangai Dream Lasers analog modulated). In case of the ramp stimulation protocol, light power changed from 0 to 100 mW/mm². Recording sessions lasted between 1 and 3 min (10–30 s laser stimulation). Yellow laser (595 nm, Shangai Dream Lasers) was used to stimulate CamKII-Halorhodopsin injected animals. Green laser (535 nm, Shangai Dream Lasers) was used to stimulate Arch-injected animals. Continuous light-stimulation (10–40 s) was used to inhibit CamKII-Halorhodopsin- and Arch-injected animals. Laser power at the tip of the fibers connected to either yellow or green laser was around 40 mW/mm². In cases where we used yellow or green light to inhibit the somas of the pyramidal or OLM$^{\alpha 2}$ in anesthetized animals, continuous light stimulation was used for 10 or 40 s. Note that we have a-priori tested the used light powers and duration, in order to avoid potential optogenetics-related artifacts[21,53]. In average, 10 sessions were recorded per each animal. Atropine sulfate (ATSO$_4$, 50 mg/kg) and scopolamine (5 mg/kg) were administrated i.p. and recordings were performed 10 min after administration.

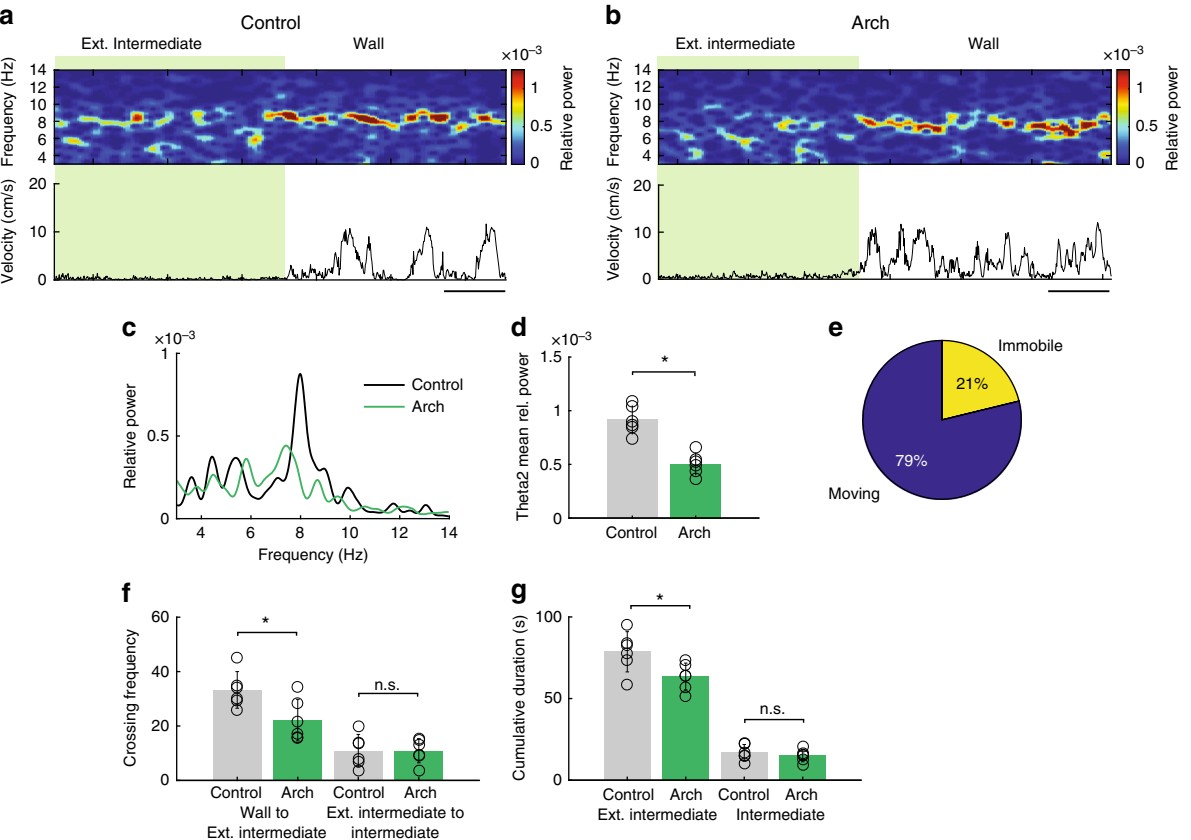

**Fig. 10** OLM$^{\alpha 2}$ cell inhibition decreases the power of theta2 in immobile animals. **a, b** Spectrograms showing decreased power of naturally occurring theta2 in immobile Chrna2/Arch animals when compared to the Control animals in the Extended Intermediate zone. In the Wall zone, only movement-related theta1 was detected. In this experiment, the green light was applied in a closed-loop manner when immobility and theta2 were both detected (see Online Methods). Scale bars: 5 s. **c, d** Power spectrum and group statistics showed a significant decrease of immobility-related theta2 in Chrna2-Cre/Arch animals ($n = 6$ per group, $*p < 0.001$, repeated-measures ANOVA). **e** Average percentage of detected movement and immobility in Chrna2/Arch animals in the Extended Intermediate zone. **f** Mean crossing frequency in Control and Chrna2/Arch animals from Wall to Extended Intermediate ($n = 6$ per group, $*p < 0.001$, $t$-test, n.s. = not significant). **g** Chrna2-Cre/Arch animals spent significantly less time exploring the Extended Intermediate zone ($n = 6$ per group, $*p < 0.001$, $t$-test, n.s. = not significant). Error bars in all bar graphs represent SEM

**In vitro electrophysiology**. Horizontal hippocampal slices from Chrna2-Cre/R26$^{tom}$ and C57/BL6 mice were obtained as after brains from 3–6 week-old mice were removed after decapitation and placed in ice-cold artificial cerebrospinal fluid (ACSF)/sucrose solution (in mM: KCl, 2.49; NaH$_2$PO$_4$, 1.43; NaHCO$_3$, 26; glucose, 10; sucrose, 252; CaCl$_2$, 1; MgCl$_2$, 4)[54]. Horizontal slices (400 µm) were collected on a vibratome and transferred to a chamber filled with recording ACSF (in mM: NaCl, 124; KCl, 3.5; NaH$_2$PO$_4$, 1.25; MgCl$_2$, 1.5; CaCl$_2$, 1.5; NaHCO$_3$, 30; glucose, 10), continuously bubbled with 95% O$_2$ and 5% CO$_2$. Slices were transferred to an upright microscope and the perfusate was maintained ~30 °C by an in-line heater. Recording pipettes were filled either with ACSF. Chrna2+ cells were visually identified (tomato fluorescence) and targeted for cell attached recordings.

**Electrode implantation for freely moving experiments**. Neuronexus probes (Neuronexus) or custom-made tungsten microwire arrays were implanted during stereotaxic surgery in mice anesthetized with isoflurane (0.5–1%) and oxygen. Animals were monitored for vital signs and hydration, and heated by a pad mounted in a stereotaxic apparatus. The head was shaved and cleaned with iodine solution and local lidocaine was applied above the periosteum. The eyes were maintained moist using a saline-based eye gel. The skull was exposed by a mid-sagittal incision, and the bone surface was brushed with a solution of saline and peroxide (3%). Craniotomies were drilled for dHipp (ML: 1.2, AP: 2) and vHipp (ML: 3.8, AP: 3.2) recordings. Stainless steel screws placed in the back of the skull served as ground and reference. In some animals recordings were made in dorsal and ventral hippocampus with Neuronexus probes and tungsten wires. For that purposes, the Intan RHD2132 16-channel amplifier board was modified by removing the soldering of two extra inputs from the amplifier and attached to the micro male pin connectors. The tungsten wires were aligned 100 µm away from each other and attached to micro female pins connectors. Electrodes were lowered by a hydraulic micromanipulator (~200 µm/min) and fixed by dental cement. Single-shank, 16-channel probes were positioned in the right vHipp and an optical

fiber was positioned ~1 mm above the Neuronexus probe; both were secured with gentle application of dental cement. Note that we stimulated OLM$^{\alpha 2}$ cells bilaterally (bilateral viral-vector injections and optical fibers bilaterally implanted). Finally, the skin was sealed around the plastic connector and the animal was placed in a new cage and maintained warm until it woke up. An analgesic (Vetergesic, 0.5 mg/kg intraperitoneal every 24 h) was administered for 2–3 days. Electrophysiological recordings started 5–7 days after surgery. After behavioral experiments, some animals were used to test the effects of optogenetic stimulation under ketamine and urethane anesthesia.

**Behavioral protocol**. After a 5–7 days recovery period, animals were food restricted to 85% body weight and habituated to the recording setup (7 daily sessions, 20 min each). Mice expressing either ChR2/YFP of control viruses were given blindly to the experimenters. Animals were then habituated to a new arena ("familiar arena" with same dimensions as home cages, 20 cm × 15 cm) in darkness with pellets available for foraging for 5 sessions of 10 min. In the next day, animals were exposed to a predator (cat) smell test for 10 min. The test was carried out at 20 lux in order to reduce the increase of theta power due to the anxiogenic effect of light[10] and to evaluate the effect of innate responses to predator smell (cat hair was placed in the center of the field in a petri dish sealed with a plastic mesh with fine apertures). The hair was taken from a verified hunter male cat that was fed frequently by hunting, a behavioral pattern that may account for the presence of compounds of prey body metabolism that is detected by rodents and suggested to trigger aversion circuits[55]. We focused in this test because of the known importance of predator smell in the induction of type 2 theta activity[4].

The closed loop laser-control for freely moving experiments was implemented in Matlab. Images of the complete arena were captured using a CMOS camera (Point Grey Flea 3). Mouse was detected using thresholding (Matlab Image Processing Toolbox). Animals were stimulated (16 Hz sinusoidal light protocol, ~4 mW laser power) when the body center point passed from the first zone (Wall) into

the Intermediate zone (surrounding the petri dish). The behavioral output (time spent) was evaluated by scoring the time inside the respective zone. The crossing frequency was determined by counting the number of animals' crossing to the neighboring zone. Electrophysiological data obtained during experiments were divided into segments by fast Fourier transform (Hamming window of 3-s length and 2.5-s overlap). Each segment was assigned to the position and velocity given by the image analysis. Figures 7d–g and 8e–h show the average power of time segments where the mice moved in the respective zones. The same groups of animals that were exposed to the predator odor test in moving animals were after 2 weeks re-exposed to this test in immobile animals.

**Olfactory test**. The same group of mice that underwent predator smell test could explore a fresh new cage (similar to the home cage, after 2 weeks) for 5 min. After a habituation period, animals were transferred to a cage with equal dimensions, but this time containing scented test stimulus placed in a small dish (good smell-peanut butter or bad smell-2-methylbutyric acid). Behavior was recorded for 2 min while the animals were optogenetically stimulated with 16 Hz sinusoidal light once they entered the zone where stimulus was placed[56].

**Theta-speed relationship evaluation**. Theta1 and induced theta2—speed relationship was evaluated in animals on a treadmill of variable speed. Animals were recorded from the state of immobility to progressively increasing speeds between 5 and 25 cm/s. The exact speed was measured using an optical switch adapted to the treadmill motor shaft. Note that similar relationship between both types of theta and speed was observed in the predatory smell test (Figs. 7d, f and 8e, g). In the Wall zone, animals moved with an average speed of ~7 cm/s, displaying a single theta peak at ~7.5 Hz (theta1). When approaching the Intermediate and Center zones, animals were moving with an average speed of ~25 cm/s. In this case, as reported before[7] and observed in the treadmill, theta1 ~9.5 Hz was detected. However, we simultaneously detected a prominent, lower frequency (~8 Hz) theta2 peak. Since the peak frequency of both rhythms correlated with running speed, different filter cutoff frequencies were used for different speeds (6–7.5 Hz for theta2 and 7.5–9 Hz for theta1 when animals run at 10 cm/s speed (Fig. 4); and 6–8 Hz for theta2 and 8–10 Hz for theta1 when animals run at ~25 cm/s (Fig. 7f, g)).

**Real-time theta2 detection**. Custom-made real-time theta2 detection software was used to detect theta2 in Control (Chrna2-Cre animals injected with eYFP control virus) and Chrna2-Cre/Arch. A baseline signal of 30 s length was recorded during freely moving mouse activity in the Wall zone. The mean baseline theta2 band (6–8 Hz) was computed from the averaged baseline signal. The closed-loop routine was implemented on top of the Intan RHD2000 Matlab interface, where amplifier data was continuously received in chunks of 1/8 s, followed by the computation of the power spectral density (PSD). Simultaneously, a video stream from the complete arena was captured using a CMOS camera and the centroid of the mouse was determined from the detected mouse shape (Matlab Image Acquisition and Image Processing Toolbox). The closed-loop routine than activated the laser output if (a) the theta2 band power in the acquired PSD was higher than the mean baseline theta2 power times a threshold and (b) the velocity of the centroid position change was greater than 5 cm/s (for movement) or less than 5 cm/s (for immobility) during the time period of the observed data chunk. The threshold between baseline and theta2 activity was determined from previous recordings from the arena and was set to 2.5. The accuracy of the detection algorithm was validated a posteriori using the full electrophysiological recording as well as the position data from the recorded video.

**Data analysis**. PSDs from selected recording sites were sessions averaged, producing a mean PSD for each animal, which counted as a single sample for the statistical analysis (sample size is equal to the number of animals). To allow merging of data from different animals, PSD values were normalized (PSDs were divided by the total power between 0 and 14 Hz before light stimulation). PSDs for each condition (pre, during, and after light) were calculated for contiguous 10/20/40-s (anesthetized) or 60-s (treadmill) LFP segments. Note that for calculating relative theta power from all hippocampal layers (anesthetized animals), only the channels where theta activity was induced during light stimulation were included in the statistical analysis (for pre, during, and after light conditions). For calculating averaged PSD in vHipp, we pulled together the data of 6 animals with electrodes implanted only in vHipp and the vHipp data of 6 animals with electrodes implanted in both vHipp and dHipp.

Position of the electrodes and the location of individual contacts of the 16-site linear silicon probe in relation to the hippocampal layers were identified histologically and by electrophysiological signatures of hippocampal SP, such as spiking activity and ripples appearance.

Phase–amplitude cross-frequency coupling was computed by means of the modulation index[24]. We used the *eegfilt* function (EEGLAB toolbox, http://sccn.ucsd.edu/eeglab/) to filter all signals. For comodulation maps, we bandpass filtered signals using 10-Hz windows and 2.5-Hz steps for the amplitude frequencies, and 0.5-Hz windows at 0.25-Hz steps for the phase frequencies. For computing the mean theta–gamma modulation index, we obtained a single modulation index for

each animal by filtering the LFP recorded from a contact in SR between 3 and 10 Hz for theta, and 30 and 80 Hz for gamma.

For unit separation, signals were bandpass filtered (500–5000 Hz) and AP automatically detected and clustered using the wave-clus software[57] with standard settings (www.vis.caltech.edu/~rodri/Wave_clus/). Isolated units were separated into regular spiking cells (RS) and interneurons (IN) based on mean firing frequency (RS < 10 spikes/s and IN > 10 spikes/s). The segregation of RS and IN was further shown by differences in AP waveforms: RS displayed mean AP half width of $0.36 \pm 0.02$ ms and IN of $0.23 \pm 0.008$ ms ($n = 40$, $p < 0.0001$, $t$ test). In addition, the ratio between the amplitude of the peak and the adjacent trough was equal to $12.3 \pm 1.9$ for RS and $1.4 \pm 0.1$ for IN. Theta firing phase preference for isolated units was calculated by bandpass filtering (3–8 Hz) the LFP from the middle channel of the probe. The instantaneous phase ($\phi(t)$) of the theta-filtered signal was calculated using the analytical representation of the signal based on the Hilbert's transform (*hilbert* function in Matlab, Signal Processing Toolbox). Each spike time was associated with a phase value obtained from $\phi(t)$. The strength of theta-phase coupling (length of the mean resultant vector, $|R|$) was obtained with the function *circ_r* from the Circular Statistics Toolbox for Matlab.

Unless otherwise stated, theta data are reported as mean ± SD and an appropriate statistical test was used (stated in figure Legends). Normal distribution of the data was tested using Kolmogorov–Smirnov test. In case of one-way repeated-measures ANOVA, Bonferroni post-hoc test was used.

**Computer simulations**. All simulations were performed using the Neuron Simulation Environment[58]. We modeled a simplified CA1 PC model that consisted of a 150 μm long distal dendrite receiving excitatory synapses from the temporammonic and Schaffer collateral pathways (modeled as an alpha function with time constant equal to 0.1 ms, 0.05 μS conductance, and 0 mV reversal potential) and the distal dendrite and soma receiving inhibitory inputs (0.3 ms time constant, −75 mV reversal potential, and 0.05 μS conductance) from OLM and basket cells, respectively. Conductances (delayed rectifier and sodium current, delayed rectifier and A-type potassium currents, $Im$ and $Ih$) were adapted from a previous CA1 PC model[59]. The model was implemented with delayed rectifier potassium and sodium conductances across soma and dendrites, an A-type and $Im$ currents in the soma and $Ih$ in the soma and distal dendrite ($Ih$ conductances were equal to 0.01 and 0.01 μS for distal dendrite and soma, respectively). Extracellular potentials were modeled as a point source using the LFPsim Neuron plugin[60].

**Cell counting, imaging, and CLARITY**. For counting Chrna2+ cells along the entire dorsoventral axis, we performed both manual cell counting in slices (30 μm coronal sections, $n = 3$ mice, 35 sections/animal) and 3D-counting using CLARITY method. Slices were imaged using Zeiss LSM 510 Meta confocal microscope.

Clearing was performed according to the CLARITY protocol (12). Mice were perfused and the hippocampi dissected before being fixed in 4% paraformaldehyde for 24 h. Following fixation, tissues were incubated at 4 °C for 3 days in 10 ml hydrogel monomer solution (Acrylamide 4%, VA-044 initiator 0.25%, 1× PBS, 4% PFA, dH₂O). Prior to polymerization, samples were degassed in a desiccation chamber and air in the head space was replaced with nitrogen. Polymerization occurred in an incubator shaker for 3 h at 37 °C. Samples were cleared at 37 °C for 2 weeks and at 45 °C for a further week in clearing solution (200 mM Boric acid, 4% Sodium Dodecyl Sulfate, dH₂O, NaOH; pH 8.5) which was replaced every 3rd day. Samples were washed twice in 1× PBST (0.1% Triton X-100) for 24 h to remove SDS from the tissue. Clear samples were refractive-index matched through three sequential 1-day incubations in 20, 40, and 63% 2,2′-thiodiethanol (TDE, Sigma-Aldrich) in 1× PBS solution. For somatostatin immunohistochemistry, following antibodies were used: SOM antibody 1:150 (MAB354, Anti-SOM Antibody, clone YC7 [Merck Millipore Corporation]). Hippocampi were imaged using a Zeiss Light Sheet Z.1 (5×/0.16 objective) and image tiles were stitched in Arivis Vision4D. Volume rendering, soma detection and counting were performed in Imaris 8.2 (Bitplane). Density plots were obtained using a custom-made Matlab software.

**Code availability**. All custom-made software for signal analysis and real-time behavioral control will be available under https://github.com/pavolbauer.

## Data availability

All data that support the findings of this study are available from the corresponding authors upon request.

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

## Acknowledgements

We thank Brian Bland, one of the discoverers of type 2 theta activity, for his insightful comments on a previous version of this manuscript; Stefan Remy for helpful discussions; George Nascimento for technical support; Steven Edwards for help with the CLARITY method; Davi Drieskens for help with illustrations; Cecilia Yates for kindly donating hair from a verified hunter cat; B. Lea and L. Hugo for their continuous motivation. This work was supported by the Kjell and Märta Beijers Foundation, The Swedish Research Council, the Swedish Brain Foundation, and the Brazilian agency CAPES (CAPES/ STINT program). S.M. was supported by SOEB foundation. C.E.R. was supported by SSMF foundation.

## Author contributions

S.M., C.E.R., R.L., and A.T. designed anesthetized experiments and analyzed the data; S.M., C.E.R., S.P., and R.L. conducted anesthetized experiments; S.M. and C.E.R. designed, conducted, and analyzed freely moving experiments with inputs from R.L. and K.K.; P.B. and S.M. wrote custom-made software for real-time behavioral control and analysis; R.L. carried out the computer simulations. S.S. performed anatomy experiments with inputs from C.E.R. and S.M.; S.M., S.S., and R.L. performed revision experiments; S.M., C.E.R., A.T., K.K., and R.L. conceptualized and wrote the paper with contribution from other authors.

## Additional information

**Competing interests:** The authors declare no competing interests.

