## [Peer Review File · Nature Communications]

Reviewers' comments:

Reviewer #1 (Remarks to the Author):

Concerning the rebuttal to the previous reviewers asking that a speed-resolved analysis should be done to dissociate the appearance of theta 2 and the speed in the anxiety task (analysed in accordance to Ahmed and Mehta 2012...

The authors have performed this analysis and the fig. supp 19 b lower panel does show that there are two theta bands that are velocity resolved and that these could be theta 1 and 2. However, the bins vs mean speed results are difficult to interpret. For example, in the ChR2 stims (supp19a), the lower panel is confusing since there is significantly less bins than the controls (does this mean that there are less segments that can be binned, maybe less segments for movements?) while it's expected that there should be more movements (since there could be less anxiety as the authors suggest). Why are the number of bins significantly different between the upper and lower panel?

Other minor comments:

1) The premise that type 1 and 2 theta can be separated into a component that is sensitive to anesthesia (ie type 2) and one that is insensitive (ie type 1) is interesting and has been proposed for a long-time in the theta literature. However, it's difficult to assert that theta 1 and 2 can be dissociated between dorsal theta 1 and ventral theta 2. For example, in the Vandercasteele et al., 2014 paper, optogenetic stimulation of ACh neurons in anesthetized animals produced a strong and clear increase in theta power in the DORSAL hippocampus (through a local increase in theta power in the LM layer). Therefore, these results suggest that ACh release can also generate type 2 theta dorsally, and not only ventrally. Therefore, this clear dorsal vs ventral type 1 and type 2 theta dichotomy may not be true and should be taken in consideration in the general discussion and supp fig. 22 could be removed.

2) Line 300-303. There is evidence in the publication cited that optogenetic stimulation of MS glutamate neurons can increase the frequency of theta but there is no reason to believe that it 'accounts' for type 1 theta or that it 'is' the generator for type 1 theta. It's more likely that type 1 is largely, or predominantly, supported by MS GABAergic neurons (Bender et al., 2015) and that MS glutamate cells only 'contributes' 'modulate' type 1 theta. Therefore, the authors should change the word 'account' if they mean that it 'explains' theta 1 theta, as it does not.

3) As for the authors' contention that ACh is associated to type II theta, this conclusion stems from the data presented in figure 1 where optogenetic stimulation of O-LM neurons in the hippocampus reduces the frequency of theta when locomotor speed is fixed (during running on a treadmill). However, this is the only result showing the role of ACh in type II theta. The effects during the predator task are not explicitly shown to be ACh dependent. In fact, there are two main issues with these results. First, it's unclear how the activation of O-LM interneurons and corresponding slowing down of theta frequency to type-2-like theta can be blocked by the muscarinic receptor antagonist. This is hard to explain at the circuit level. The authors propose that it's the ACh that activates O-LM interneurons and not the other way around! Second, a recent report (Carpenter et al., 2017) showed that DREADD activation of medial septum ACh neurons produces a slowing down of theta without changing locomotion, in agreement with the present results. However, behaviorally, activating ACh neurons lead to an increase in anxiety-like behavior in an open field (decrease in movements and less time spent in the center of the arena) which is the opposite of what is expected from the present study (activation of O-LM

interneurons increases crossing frequency into the intermediate or central zone containing the odor). How can the results from Carpenter et al., 2017 and the present results be reconciled if ACh is equally involved? While ACh may well be involved in the slowing down of theta frequency, it has not been convincingly shown that it is involved in the behavior shown by the authors in their behavioral task. At the least, this should be discussed by the authors and again supp Fig. 22 should be altered or removed.

Reviewer #1 (Remarks to the Author):

The authors have performed this analysis and the fig. supp 19 b lower panel does show that there are two theta bands that are velocity resolved and that these could be theta 1 and 2. However, the bins vs mean speed results are difficult to interpret. For example, in the Chr2 stims (supp19a), the lower panel is confusing since there is significantly less bins than the controls (does this mean that there are less segments that can be binned, maybe less segments for movements?) while its expected that there should be more movements (since there could be less anxiety as the authors suggest). Why are the number of bins significantly different between the upper and lower panel?

We thank the reviewer for his/her further work on our submission.

We realize we could be clearer. Please note that in Supp. Figure 19a (now current Supp Fig 17a), the division between the upper and lower panels is not between control animals and animals stimulated by Chr2 activation. Rather, the upper panel shows the distribution of speeds for when the Chr2 animal was in the Wall zone, while the lower panel shows the speed distribution for when the Chr2 animal was in the Intermediate zone. Therefore, the number of bins does not need to be equal in the upper and lower panels, since it reflects the time the animal spent on each region of the arena. Because the animal spent less time in the Intermediate zone, there are fewer bins in the lower panel. In the newly revised version, we have edited the Supp. Figure 17a legend to make this clearer.

Other minor comments:

1) The premise that type 1 and 2 theta can be separated into a component that is sensitive to anesthesia (ie type 2) and one that is insensitive (ie type 1) is interesting and has been proposed for a long-time in the theta literature. However, its difficult to assert that theta 1 and 2 can be dissociated between dorsal theta 1 and ventral theta 2. For example, in the Vandercasteele et al., 2014 paper, optogenetic stimulation of ACh neurons in anesthetized animals produced a strong and clear increase in theta power in the DORSAL hippocampus (through a local increase in theta power in the LM layer). Therefore, these results suggest that ACh release can also generate type 2 theta dorsally, and not only ventrally. Therefore, this clear dorsal vs ventral type 1 and type 2 theta dichotomy may not true and should be taken in consideration in the general discussion and supp fig. 22 could be removed.

This is an interesting remark and we agree that it should be integrated into the discussion. It is correct that in Vandercasteele et al., 2014 paper the authors report theta increase in the dorsal hippocampus upon local cholinergic stimulation in the medial septum. This does not exclude the possibility that the cholinergic-driven theta emerges first in the ventral hippocampus and spreads to its dorsal counterpart – as our data suggest. That possibility is also coherent with the significantly denser cholinergic innervation from the medial septum in the ventral than in the dorsal hippocampus (<http://mouse.brain-map.org/>). We have now updated the discussion (*Discussion, paragraph 4*) to include this comment.

Regarding the Supplementary Figure 22, we agree with the reviewer that this model is not complete as the field still lacks conclusive evidence regarding MS GABAergic innervation of hippocampal interneurons, as well the local connectivity in MS. Thus, we followed the reviewer's suggestion and removed this figure.

2) Line 300-303. There is evidence in the publication cited that optogenetic stimulation of MS glutamate neurons can increase the frequency of theta but there is no reason to believe that it 'accounts' for type 1 theta or that it 'is' the generator for type 1 theta. Its more likely that type 1 is largely, or predominantly, supported by MS GABAergic neurons (Bender et al., 2015) and that MS glutamate cells only 'contributes' 'modulate' type 1 theta. Therefore, the authors should change the word 'account' if they mean that it 'explains' theta 1 theta, as it does not.

We have replaced the word "accounts" with "contributes" and now also cite the published evidence that MS GABAergic neurons are involved in type 1 theta rhythmogenesis (*Discussion, paragraph 6*).

3) As for the authors contention that ACh is associated to type II theta, this conclusion stems from the data presented in figure 1 where optogenetic stimulation of O-LM neurons in Vhippocampus reduces the frequency of theta when locomotor speed is fixed (during running on a treadmill). However, this is the only result showing the role of ACh in type II theta. The effects during the predator task is not explicitly shown to be ACh dependent. In fact, there are two main issues with these results. First, its unclear how the activation of O-LM interneurons and corresponding slowing down of theta frequency to type-2 -like theta can be blocked by the muscarinic receptor antagonist. This is hard to explain at the circuit level. The authors propose that it's the ACh that activates O-LM interneurons and not the other way around! Second, a recent report (Carpenter et al., 2017) showed that DREADD activation of medial septum ACh neurons produces a slowing down of theta without changing locomotion, in agreement with the present results. However, behaviorally, activating ACh neurons lead to an increase in anxiety-like behavior in an open field (decrease in movements and less time spend in the center of the arena) which is the opposite of what is expected from the present study (activation of O-LM interneurons increases crossing frequency into the intermediate or central zone containing the odor). How can the results from Carpenter et al., 2017 and the present results be reconciled if ACh is equally involved? While ACh may well be involved in the slowing down of theta frequency, it has not been convincingly shown that it is involved in the behavior shown by the authors in their behavioral task. At the least, this should be discussed by the authors and again supp Fig. 22 should be altered or removed.

The points the reviewer is raising are important and we agree that they should be discussed. We did not apply atropine in the same manner we did in the treadmill experiments since atropine was previously shown to affect respiration and animals' baseline anxiety, which could have biased our behavioral experiments. Further, we observed a similar, slower theta peak during the predator odor test and during the treadmill experiments, we therefore assumed that the cholinergic blockade would abolish type 2 theta also in this condition. We do agree with the reviewer that the role of MS cholinergic (and other) inputs onto OLM cells should be investigated in detail in future studies. As mentioned in our response to point 1, we now state the importance of future studies addressing these questions in the *Discussion, paragraph 6*.

The results presented in Carpenter et al., 2017 (that we now cite) are interesting and also in line with previous studies that focused on the open-field test (using light in the center of the arena) which showed that the amplitude of theta activity in vHipp is higher when animals are in the safe zone (borders) of an open field (Adhikari et al, 2010). Thus, we believe that the generation of theta2 by OLM^{α2} cell stimulation could be related to the perception of safety and associated with the arousal state of an animal. This would imply that OLM^{α2} driven theta2 activity could contribute to different

behavioral outputs, depending on the behavioral context in which the animal is involved. Future studies should address this question in detail – we state this in the *Discussion, paragraph 8*.